# Development of a New Methodology for Dearomative Borylation of Coumarins and Chromenes and Its Applications to Synthesize Boron-Containing Retinoids

**DOI:** 10.3390/molecules28031052

**Published:** 2023-01-20

**Authors:** Bhaskar C. Das, Pratik Yadav, Sasmita Das, Mariko Saito, Todd Evans

**Affiliations:** 1Arnold and Marie Schwartz College of Pharmacy and Health Sciences, Long Island University, Brooklyn, NY 11201, USA; 2Division of Nephrology, Department of Medicine, Icahn School of Medicine at Mount Sinai, New York, NY 10029, USA; 3Nathan S. Kline Institute for Psychiatric Research, Orangeburg, NY 10962, USA; 4Department of Surgery, Weill Cornell Medical College of Cornell University, New York, NY 10065, USA

**Keywords:** borylation, coumarins, chromenes, retinoinds, conjugate addition, oxacycles

## Abstract

Dearomative borylation of coumarins and chromenes via conjugate addition represents a relatively unexplored and challenging task. To address this issue, herein, we report a new and general copper (I) catalyzed dearomative borylation process to synthesize boron-containing oxacycles. In this report, the borylation of coumarins, chromones, and chromenes comprising functional groups, such as esters, nitriles, carbonyls, and amides, has been achieved. In addition, the method generates different classes of potential boron-based retinoids, including the ones with oxadiazole and anthocyanin motifs. The borylated oxacycles can serve as suitable intermediates to generate a library of compounds.

## 1. Introduction

Retinoic acid (RA) signaling is among the most significant biological pathways facilitated at its core by the interaction of RA and nuclear receptors that control gene expression [1]. Retinoic acid derivatives (Retinoids) govern numerous cellular phenomena, including inhibition of proliferation, induction of cell differentiation, regulation of apoptosis, and inhibition of inflammation [2,3,4]. However, due to the complexity of ligands and receptors and involvement in several biological processes, RA can cause side effects. Moreover, RA suffers from poor solubility, photosensitivity, and increased catabolism during intravenous administration. All these limitations encourage the development of novel and better-tolerated synthetic retinoids. Previously, our group discovered that the toxicity of retinoids might be significantly suppressed by the incorporation of an oxygen-heterocyclic ring in the hydrophobic part of RA [5].

Oxygen-heterocycles are integral parts of various natural products and bioactive molecules [6]. Among them, coumarins and chromenes belong to a privileged class of heterocycles due to their inherent biological as well as photo-physical properties [7]. Natural as well as synthetic coumarins are well known for their broad spectrum of pharmacological activities, including acting as anti-microbials [8], antioxidants, anti-inflammatory agents [9], anti-depressants [10], anti-tumor agents [11], anti-asthmatics [12], antivirals (with anti-HIV) [13], anti-nociceptives [9], and anti-coagulants [14]. Similarly, chromenes are widespread in nature and have displayed excellent biological activities [7].

Meanwhile, boron-based molecular architects have recently emerged as potential pharmaceutical agents and probes due to their interesting and broad range of activities [15]. In addition, organoboron reagents are suitable for various organic transformations [16,17,18]. Several boron-based scaffolds are under clinical trials [19], and Bortezomib (Velcade), Tavaborole (Kerydin), Crisaborole (Eurisa), Vabomere, and Epetraborole are FDA-approved boron-based marketed drugs [20,21,22]. Vacant p-orbitals of boron make boron-based scaffolds prone to accepting electrons from electron-donating atoms as well as functional groups. This governs the strong interaction of boron chemicals at the active sites of receptors. The boron-based molecular architects also possess a tendency to form hydrogen bond networks to govern several biological processes and play a major role in drug–receptor interaction. The unique nature of boron-based compounds makes them potential therapeutic agents for the development of new drug candidates [15]. Therefore, we rationalized that the amalgamation of boron and coumarins/chromenes could lead to a new class of therapeutic agents with enhanced pharmaceutical properties, wherein boron-based heterocyclics could be potential hydrophobic substitutes for RA [5,23]. In addition, boron-containing oxacycles can be efficiently used for various synthetic transformations to generate a library of biologically active compounds. Based on these concepts, we previously synthesized a novel RARα agonist [5], **BD4** (Figure 1), which reduces proteinuria and recovers kidney injury, and mice administrated with **BD4** did not develop any obvious toxicity or side effect. Therefore, **BD4** appeared to be a new proof-of-concept RARα agonist, which could be used as a potential therapy for patients with kidney diseases, such as HIV-associated nephropathy (HIVAN). However, **BD4** has low solubility and relatively low affinity. Therefore, we planned to use **BD4** as our starting point to develop better compounds for lead optimization.

Based on our study, we realized that boron could be directly added to the endo/exo-cyclic double bond of a chromene motif (Figure 1) to generate **BD4a** or **BD4b**. The literature revealed that initially, Blum and co-workers synthesized borylated isocoumarins from esters via electrophilic oxyboration [24]. Later, Yudin et al. achieved the synthesis of boryliminocoumarin from borylketenimines [25], although these methods were limited to certain building blocks. On the other hand, the metal-catalyzed conjugate addition of diboranes remained a valuable tool for inserting boron into unsaturated molecular motifs [26,27,28,29,30,31,32]. In recent years, copper-catalyzed conjugate addition reactions [32] of boron have emerged as a viable method to construct borylated heterocycles [33,34]. Among them, the dearomative borylation strategy of indoles [35,36,37], pyrroles [37,38], pyridines [39], and 4-quinolinols [40] have been efficiently utilized to construct borylated analogs of these biologically significant frameworks. Conjugate borylation of substituted esters and ketones has also been studied, which has demonstrated various challenges in the efficient conversion into corresponding borylated products [41,42]. However, to the best of our knowledge, dearomative borylation of coumarins and chromenes bearing various pharmaceutically relevant functional groups, including esters, amides, nitriles, carbonyls, and nitrogen heterocycles, has not yet been achieved. In addition, the application of these reported conjugate borylation strategies on medicinally important scaffolds has remained an unexplored task.

As a part of our ongoing effort on the development of boron-based therapeutic agents [43,44,45,46], we envisioned that under suitable reaction conditions, copper-catalyzed conjugate addition onto coumarins/chromones could result in borylated coumarins/chromones via a dearomative mechanism and that we could introduce these compounds to our RA backbone to substitute the hydrophobic part. To explore and expand our hypothesis, we herein report a conjugate addition of diboranes onto coumarins and chromones under simple and efficient conditions. The strategy was further utilized for the synthesis of biologically relevant novel boron-based retinoids. In addition, these borylated scaffolds were also used as synthetic precursors for diversified biological active compounds.

## 2. Results and Discussion

### 2.1. Reaction Optimization

To initiate an investigation for the dearomative borylation of coumarins, ethyl coumarin-3-carboxylate and Bis(pinacolato)diboron [B_2_(pin)_2_] were chosen as model substrates. To carry out the transformation, various reaction conditions were screened (Table 1). The planned reaction was realized very challenging due to the following: 1. the presence of a highly crowded tri-substituted alkene reaction site; 2. a competitive deconstruction of coumarins under copper (I) catalyzed conditions [47]. Our initial efforts yielded the conversion of coumarins into salicylaldehyde **2a′** under various conditions (Table 1). After screening numerous reaction conditions, we successfully suppressed the formation of salicylaldehyde. Initial results showed that copper (I) in combination with phosphine ligand and NaO*^t^*Bu with methanol in THF could furnish product Ethyl 2-oxo-4-(4,4,5,5-tetramethyl-1,3,2-dioxaborolan-2-yl)chromane-3-carboxylate (**2a**). Copper (I) chloride (CuCl) was found to be a better copper (I) source (Table 1, entries **1**, **2**) in comparison to Copper (I) iodide (CuI). In search of better reaction conditions for this transformation, we screened various phosphines (Table 1, entries **1**–**7**). Different biphenyl derived phosphine served the desired product **2a,** but ligands, such as triphenylphsphine (PPh_3_), tricyclohexylphosphine (PCy_3_), and tributylphosphine (P*n*Bu_3_), were found better with respective yields of 60%, 82% and 75% for **2a**. Having a suitable phosphine source, we screened various bases and additives to compare their effect on the outcome of the reaction, and the results are summarized in Table 1, entries **8**–**12**. Based on the optimization of the reaction, CuCl was found as an efficient copper (I) source, in combination with PCy_3_, NaO*^t^*Bu, and MeOH in THF at room temperature.

### 2.2. Substrate Scope of Dearomative Borylation

With the optimized reaction conditions in hand, we evaluated the substrate scope of this transformation (Figure 2). Substrate, methyl coumarin-3-carboxylate with B_2_(pin)_2_ under the developed reaction condition gave the desired product **2b** in 80% yield. Encouraged by these results, we next turned our focus to gauging the effect of an ester group on the C-3 position of coumarins with this protocol. Therefore, a reaction of coumarin was performed under the above reaction conditions for **2a** synthesis. We were able to obtain the desired product **2c** with unsubstituted coumarin as well in 78%. Next, the outcome of this dearomative borylation process was investigated with differently substituted C-3 positions. The reaction was also found efficient for the synthesis of reactants bearing nitriles and carbonyl functional groups to provide compounds **2d** and **2e** in good yields. Due to the excellent biological activities of chlorochromenes [43,44,45,46] and to evaluate the effect of substituents, we drew our attention toward the synthesis of **2f** and **2g**. Halogen-bearing substrates are usually found interfering and intolerant of Miyaura borylation reactions. However, in contrast, under these dearomative borylation conditions, these groups were well tolerated to yield **2f** and **2g**. To check the outcome of the reaction with an amide containing coumarins, a reaction of 3-acetamidocoumarin was treated with B_2_(pin)_2_ under the developed conditions, which resulted in the formation of **2h** in moderate yield (53%) via in-situ deacetylation-borylation sequence. Our efforts to convert coumarin-3-carboxylic acid into corresponding borylated product **2i** did not succeed.

We next explored the scope of this dearomative borylation strategy with other classes of oxacycles. We treated various chromenes with B_2_(pin)_2_ and bis(neopentyl glycolato)diboron [B_2_(neo)_2_] under the same catalytic conditions (Figure 3). Initially, chromenes were investigated to synthesize compound **4a**. Indeed, we successfully converted into **4a** with a good yield. We were also interested to see the effect of another boron source on the transformation. To investigate this, a reaction was performed with B_2_(neo)_2_ under similar reaction conditions. The reaction proceeded well to yield 74% of the desired compound **4b**. However, our attempt to construct compound **4c** did not result in the desired product, and the starting material remained unreacted.

### 2.3. Synthetic Applications of Borylated Coumarins

Organoboron compounds have been found as excellent reagents for numerous synthetic transformations [16]. Furthermore, their trifluroborate salts and oxaboroles have been realized to possess excellent pharmacological properties [15]. Therefore, we became interested in the synthesis of coumarin-based trifluroborate salts and oxaboroles. After the successful borylation of various functionalized coumarins and chromenes, we also wished to demonstrate that the borylated coumarins could serve as useful intermediates, including ethyl 4-hydroxy-2-oxochromane-3-carboxylate (**5b**) (Figure 4). To evaluate our hypothesis, **2a** was first converted into potassium ethyl 2-oxo-4-(trifluoro-l4-boranyl)chromane-3-carboxylate (**5a**). The reaction proceeded smoothly to provide **5a** (80% yield). In the second transformation, oxidation of **2a** in the presence of NaBO_3_ was attempted, which provided compound **5b** an 85% yield. In contrast, our attempt to prepare oxaborole (**5c**) in the presence of NaBH_4_ in MeOH at room temperature failed to give the desired product, and we could only get a starting material. Further attempts to explore the synthetic utilities of these borylated oxacycles are ongoing currently.

### 2.4. Application of Dearomative Borylation Strategy

The long-term goal of our research group is to design and synthesize boron-based biologically important scaffolds for various biological targets [48,49]. Among them, suitably functionalized oxadiazoles and chromenes were explored as potential pharmacophores. Chromene **6a**, an analog of naturally occurring anthocyanins, has been used as a novel retinoic acid receptor alpha agonist for the treatment of kidney disease [5,49]. In addition, oxadiazoles were also explored as pan-RAR inverse agonists [48]. Moreover, anthocyanins derived from conjugated boron-based scaffolds were also found as active retinoids [5,49]. We were, therefore, interested to utilize this protocol to generate a library of boron-based pharmacophores as part of our ongoing chemical biology program. To achieve the goal, we further extended the scope of this reaction for the synthesis of compounds **7a**–**c**.

A new chromene-based compound 6,8-dichloro-2-phenyl-2H-chromene-3-carbonitrile **6a** was prepared in good yields for the first time, starting from corresponding aldehyde **5d**. A **6a** was then treated with B_2_(pin)_2_ under the developed conditions to yield borylated compound **7a** with 55% yield. It was noteworthy that compounds **6b** and **6c** can also be converted into desired novel boron-based retinoids **7b** and **7c** in decent yields without any modification in the reaction conditions (Figure 5). From the initial hit **BD4** (**BT75**) as a novel RARα agonist, we can now access the new **BD4b** (**7c**) analog regioselectively. Further development of this strategy is under investigation to generate a library of these scaffolds. The methodology provides access to the differently functionalized borylated chromenes and coumarins. We note that the synthesized borylated oxacycles were observed to be prone to decomposition, making their purification a challenging task. The method also provides diastereomeric mixtures in many cases. Therefore, our group is currently working toward a suitable chiral approach to this process.

### 2.5. Proposed Mechanism of Dearomative Borylation of Oxacycles

On the basis of the literature, a mechanism was also proposed for this copper(I)-catalyzed dearomative borylation of oxacycles (Figure 6) [33,34,35,36,37]. In the first step, Cu(I)Cl would react with base (NaO*^t^*Bu) and ligand (**L**) to generate the active catalyst **A,** which subsequently would transfer B_2_pin_2_ onto the Cu(I) center to form intermediate **B**. In the next step, copper would coordinate with unsaturated oxacycles (**1**) to generate intermediate **C**. Subsequently, 3,4-addition of boron would result in a copper(I)-enolate and the formation of a C –B bond. After the formation of intermediate **D**, the proton donor (MeOH) would furnish the synthesis of desired product **2** and regenerate the active catalyst **A**.

## 3. Conclusions

In conclusion, we report here for the first time the dearomative synthesis of newboron-based oxygen heterocycles. The reaction works efficiently with various coumarin and chromenes, while functional groups, such as ester, nitrile, carbonyl amide, and chloro, were well tolerated. The developed methodology was also used for the development of boron-based new retinoids and biologically relevant scaffolds. These boron-based coumarins also serve as potential intermediates. This process provides a useful alternative to generate a library of coumarin/chromene-based molecular architects for future drug and molecular probe development. Further development of this approach and biological activities of synthesized oxacycles are currently under investigation at our laboratory.

## Data Availability

All the supporting data has been provided in Appendix A.

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
