# Peer review of "Development of a New Methodology for Dearomative Borylation of Coumarins and Chromenes and Its Applications to Synthesize Boron-Containing Retinoids"

_molecules, 2023, doi:10.3390/molecules28031052_

Round 1

Reviewer 1 Report

In this article, the authors reported the dearomative synthesis of novel boron-based oxygen heterocycles for the first time. Dearomative borylation of coumarins and chromenes via conjugate addition represents a relatively unexplored and challenging task. To address this issue, authors reported a novel and general copper (I) catalyzed dearomative borylation process to synthesize boron-containing oxacycles. The authors showed the reaction works efficiently with various coumarin and chromenes, while functional groups such as esters, nitriles, carbonyl amides, and chloro were well tolerated. This process provides a useful alternative to generate a library of coumarin/chromene-based molecular architects for future drug and molecular probe development. The developed methodology was also used to synthesize boron-based novel retinoids and biologically relevant scaffolds. These boron-based chromenes also serve as potential intermediates. This paper is well-written and has a very innovative approach to synthesizing boron-based Organoboron compounds. All newly synthesized compounds are well characterized with NMR  (1H, 13C, and HRMS). All references are well documented.

Based on the above observation, I recommend this paper to be published to Molecules. There are a few typo errors in supporting information, and some new references are recommended to be added.

 In Page 2: Our initial efforts yielded conversion of coumarins into salicylaldehyde 2a’ under various conditions (SI, Fig. 1)”.  SI, Fig. 1 should be Table 1.

Author Response

Responses to the reviewer 1’s comments

Reviewer 1:

Comments and Suggestions for Authors

In this article, the authors reported the dearomative synthesis of novel boron-based oxygen heterocycles for the first time. Dearomative borylation of coumarins and chromenes via conjugate addition represents a relatively unexplored and challenging task. To address this issue, authors reported a novel and general copper (I) catalyzed dearomative borylation process to synthesize boron-containing oxacycles. The authors showed the reaction works efficiently with various coumarin and chromenes, while functional groups such as esters, nitriles, carbonyl, amides, and chloro were well tolerated. This process provides a useful alternative to generate a library of coumarin/chromene-based molecular architects for future drug and molecular probe development. The developed methodology was also used to synthesize boron-based novel retinoids and biologically relevant scaffolds. These boron-based chromenes also serve as potential intermediates. This paper is well written and has a very innovative approach to synthesizing boron-based Organoboron compounds. All newly synthesized compounds are well characterized with NMR (1H, 13C, and HRMS). All references are well documented. Based on the above observation, I recommend this paper to be published to Molecules. There are a few typo errors in supporting information, and some new references are recommended to be added.

Response: We would like to thank reviewer for his valuable comments, appreciating our work and recommending the manuscript for the publication in Molecules. We have added some new references in manuscript.

Comment 1: In Page 2: “Our initial efforts yielded conversion of coumarins into salicylaldehyde 2a’ under various conditions (SI, Fig. 1)”. SI, Fig. 1 should be Table 1.

Response: As per the suggestion of reviewer, we have corrected it in the revised manuscript.

Fig. 1 should be Table 1.

Reviewer 2 Report

In this manuscript, the authors introduced a novel and general copper (I) catalyzed dearomative borylation process to synthesize boron-containing oxacycles in detail. On this basis, they further investigated a new methodology for dearomative borylation of coumarins and chromenes and its application in the development of boron-based novel retinoids and biologically relevant scaffolds. Meanwhile, they also investigated that these boron-based coumarins serve as potential intermediates. The manuscript is clearly and amply stated. I would suggest accepting it after the following concerns are addressed.

1.   The fourth corresponding address is not cited from authorship list.

2.   In the section of results and discussion, it is suggested to add subheadings according to the purpose of different experiments to make the context clearer.

3.   In terms of evaluating that the borylated coumarins could serve as useful intermediates, explanations of the results of scheme 4 should be supplemented.

4.   In the section of conclusion, either the sentence that “these boron-based chromenes also serve as potential intermediates” should be corrected to “these boron-based coumarins also serve as potential intermediates”, or it is suggested supplying relevant experiments to prove that these boron-based chromenes also serve as potential intermediates.

Author Response

Responses to the reviewer 2’s comments

Reviewer 2:

Comments and Suggestions for Authors

In this manuscript, the authors introduced a novel and general copper (I) catalyzed dearomative borylation process to synthesize boron-containing oxacycles in detail. On this basis, they further investigated a new methodology for dearomative borylation of coumarins and chromenes and its application in the development of boron-based novel retinoids and biologically relevant scaffolds. Meanwhile, they also investigated that these boron-based coumarins serve as potential intermediates. The manuscript is clearly and amply stated. I would suggest accepting it after the following concerns are addressed.

Response: We would like to thank reviewer for his valuable comments, appreciating our work and recommending the manuscript for the publication in Molecules.

Comment 1: The fourth corresponding address is not cited from authorship list.

Response: We would like to thank reviewer for pointing out this. We have added the fourth corresponding address in the revised manuscript.

Comment 2:  In the section of results and discussion, it is suggested to add subheadings according to the purpose of different experiments to make the context clearer.

Response:  As per the suggestion of reviewer, we have incorporated subheadings in the section of results and discussion to make the context clearer.

Comment 3:  In terms of evaluating that the borylated coumarins could serve as useful intermediates, explanations of the results of scheme 4 should be supplemented.

Response:  As per the suggestion of reviewer, we have supplemented explanations of the results of scheme 4.

Comment 4:  In the section of conclusion, either the sentence that “these boron-based chromenes also serve as potential intermediates” should be corrected to “these boron-based coumarins also serve as potential intermediates”, or it is suggested supplying relevant experiments to prove that these boron-based chromenes also serve as potential intermediates.

Response:  According to reviewer’s suggestion, we have corrected either the sentence that “these boron-based chromenes also serve as potential intermediates” should be corrected to “these boron-based coumarins also serve as potential intermediates”. Thank you for pointing this out. We also used these intermediates to develop some new compounds in the Scheme 4.

Reviewer 3 Report

1. I suggest the author give a more detail explanation in the introduction “… a clearer storyline should be introduced by choosing a suitable entry point to explain the reasons for designing such boron-containing retinoids”

2. Why used the word “nove;”, not new?

3. Some copper (I) catalyzed dearomative borylation process has been updated and highlighted, such as Chem. Commun., 2022, 58, 6653–6656; Org. Chem. Front., 2020,7, 3515-3520; New J. Chem., 2020, 44, 16265-16268; J. Org. Chem. 2019, 84, 14627−14635 and Org. Chem. Front., 2021, 8, 4554–4559

4. Source and purity of all chemicals used should be specified in the experimental section.

5. Some spelling errors in the text should be avoided.

6. I suggest the authors have a copper (I) catalyzed mechanism for this highlighted work.

7. Also compared the documents and give a Table for comparison.

Author Response

Responses to the reviewer 3’s comments

Reviewer 3:

Comments and Suggestions for Authors

Comment 1:  I suggest the author give a more detail explanation in the introduction “… a clearer storyline should be introduced by choosing a suitable entry point to explain the reasons for designing such boron-containing retinoids”.

Response:  As per the suggestion of reviewer, we have incorporated more explanation in the introduction.

Comment 2: Why used the word “nove;”, not new?

Response:  We have replaced the word “novel” with “new” in the manuscript.

Comment 3: Some copper (I) catalyzed dearomative borylation process has been updated and highlighted, such as Chem. Commun., 2022, 58, 6653–6656; Org. Chem. Front., 2020, 7, 3515-3520; New J. Chem., 2020, 44, 16265-16268; J. Org. Chem. 2019, 84, 14627−14635 and Org. Chem. Front., 2021, 8, 4554–4559.

Response: As per the suggestion of reviewer, we have incorporated the references of copper catalyzed 1,6-addition (Chem. Commun., 2022, 58, 6653–6656) and recent applications of boron reagents in synthetic organic chemistry  (Org. Chem. Front., 2020, 7, 3515-3520; Org. Chem. Front., 2021, 8, 4554–4559) in the revised manuscript.  The new references read as 15, b,c,d and 22f.

Comment 4: Source and purity of all chemicals used should be specified in the experimental section.

Response: As per the suggestion of reviewer, we have incorporated Source and purity of all chemicals used in the experimental section.

Comment 5: Some spelling errors in the text should be avoided.

Response: As per the suggestion of reviewer, we have rechecked and rectified the spelling/typo errors in the revised manuscript.

Comment 6: I suggest the authors have a copper (I) catalyzed mechanism for this highlighted work.

Response: We would like to thank the reviewer for suggesting this. As per the suggestion, we have added proposed mechanism in the revised manuscript (scheme 6).

Comment 7: Also compared the documents and give a Table for comparison.

Response: To the best of our knowledge dearomative borylation of coumarins and chromenes bearing various pharmaceutically relevant functional group including esters, amides, nitriles, carbonyls and nitrogen heterocycles has not yet been achieved. Therefore, we do not find literature to compare the documents for a Table for comparison.

However, in the introduction section we have incorporated various efforts for the synthesis of borylated heterocycles “Yudin et al. achieved synthesis of boryliminocoumarin from borylketenimines [21] although these methods are limited to certain building blocks. On the other hand, metal-catalyzed conjugate addition of diboranes remained a valuable tool to insert boron into unsaturated molecular motifs [22]. In recent years, copper catalyzed conjugate addition reactions of boron has emerged as a viable method to construct borylated heterocycles [23]. Among them dearomative borylation strategy of indoles [24], pyrroles [25], pyridines [26], and 4-quinolinols [27] have been efficiently utilized to construct borylated analogues of these biologically significant frameworks. Conjugate borylation of substituted esters and ketones has also been studied which has demonstrated various challenges in the efficient conversion into corresponding borylated products [28]”.